

# Searching for strongly coupled AdS matter with multi-trace deformations

**Luis Apolo[1]⋆, Alexandre Belin[2,3]† and Suzanne Bintanja[4]‡**

**1** Beijing Institute of Mathematical Sciences and Applications, Beijing 101408, China
**2** Dipartimento di Fisica, Università di Milano - Bicocca, I-20126 Milano, Italy
**3** INFN, Sezione di Milano-Bicocca, Piazza della Scienza 3, 20126 Milano, Italy
**4** Institute for Theoretical Physics and Δ-Institute for Theoretical Physics,
University of Amsterdam, PO Box 94485, 1090GL Amsterdam, The Netherlands

⋆ apolo@bimsa.cn , † alexandre.belin@unimib.it , ‡ s.bintanja@uva.nl

## Abstract

Holographic CFTs admit a dual emergent description in terms of semiclassical general relativity minimally coupled to matter fields. While the gravitational interactions are required to be suppressed by the Planck scale, the matter sector is allowed to interact strongly at the AdS scale. From the perspective of the dual CFT, this requires breaking large-$N$ factorization in certain sectors of the theory. Exactly marginal multi-trace deformations are capable of achieving this while still preserving a consistent large-$N$ limit. We probe the effect of these deformations on the bulk theory by computing the relevant four-point functions in conformal perturbation theory. We find a simple answer in terms of a finite sum of conformal blocks, indicating that the correlators display no bulk-point singularities. This implies that the matter of the bulk theory is made strongly coupled by boundary terms rather than local bulk interactions. Our results suggest that holographic CFTs that describe strongly coupled AdS matter must be isolated points on the CFT landscape or sit infinitely far away on the conformal manifold from conventional holographic CFTs.


# 1   Introduction

Characterizing the space of consistent theories of quantum gravity remains one of the big open problems in high energy physics. This problem can be formulated in its sharpest form for theories of quantum gravity with a negative cosmological constant, i.e. in asymptotically Anti-de Sitter (AdS) spaces, thanks to holography and the AdS/CFT correspondence [1–3]. In $AdS_{d+1}$ space, a theory of quantum gravity is consistent if it leads to boundary correlation functions that satisfy the fundamental axioms of a conformal field theory in $d$ dimensions ($CFT_d$): unitarity, causality, and crossing symmetry. A theory of quantum gravity is consistent if it complies with these conditions not just in the semiclassical expansion, but at the full nonperturbative level.

One can thus view the space of CFTs as describing the space of consistent theories of quantum gravity in AdS space. This statement is rather formal, since most CFTs will not lead to gravity duals which are accurately described at low energies by semiclassical general relativity coupled to a finite number of matter fields. CFTs that admit a dual, emergent description in terms of semiclassical general relativity minimally coupled to matter fields are very special and are often referred to as *holographic CFTs*. In AdS, one can thus reformulate the goal of characterizing the space of consistent theories of quantum gravity in terms of two questions: what are the defining properties of a holographic CFT, and what is the list of CFTs that satisfy these properties?

This way of phrasing the question is physical and practical at the same time. It is physical because it underlines the key dynamical properties that enable the emergence of a geometric description at low energies; but it is also practical, as it efficiently organizes the search. Much progress on this topic has been achieved in recent years, most prominently thanks to an application of the conformal bootstrap program [4–6] to holography, as pioneered in [7]. The starting point of this program, and the first known condition of holographic CFTs, is the large-$N$ condition. This condition introduces a parametric separation between the AdS and Planck scales that is parametrized in the CFT by the stress tensor two-point function

$$\langle TT \rangle \sim N \sim \frac{\ell_{AdS}^{d-1}}{G_N}\,, \tag{1}$$

where $N$ schematically counts the number of local degrees of freedom of the CFT, $\ell_{AdS}$ is the scale of AdS, and $G_N$ is Newton's constant. If $N$ is large, the bulk gravitational theory is semiclassical, i.e. weakly coupled, such that graviton loops are suppressed.

The second condition required by [7] (see also [8]) is large-$N$ factorization. We will review this condition in detail in section 2. In essence, it implies that correlation functions of operators with scaling dimensions $\Delta \ll N$ factorize in the large-$N$ limit, such that light operators behave as generalized free fields. In the bulk, large-$N$ factorization implies that all couplings are suppressed by the Planck scale. For example, the effective Lagrangian of a scalar field would

be

$$\mathcal{L}_{\text{scalar}} = \frac{1}{2}\partial_\mu \phi \partial^\mu \phi + m^2 \phi^2 + \tilde{g}_3 \frac{G_N^{1/2}}{\ell_{\text{AdS}}^2}\phi^3 + \tilde{g}_4 \frac{G_N}{\ell_{\text{AdS}}^2}\phi^4, \tag{2}$$

for order one values of $\tilde{g}_{3,4}$. These bulk couplings can be matched to parameters in the solution to the bootstrap equations in the $1/N$ expansion of the dual CFT [7,9]. Note that the scaling of the couplings makes the effective bulk theory tractable as it becomes free in the semiclassical ($G_N \to 0$) limit.

Large-$N$ factorization holds in canonical realizations of AdS/CFT, and follows naturally if the boundary CFT is a large-$N$ gauge theory. However, a common misconception is that large-$N$ factorization is a necessary condition for all holographic CFTs.[1] The only condition that is truly necessary is that gravity is semiclassical, i.e. weakly coupled. This can be diagnosed from the four-point function of the stress tensor in the CFT, the condition being

$$\frac{\langle T T T T \rangle_c}{\langle T T \rangle \langle T T \rangle} \sim \frac{1}{N}. \tag{3}$$

If large-$N$ factorization is imposed, this condition follows, but the converse is not true (at least it does not imply large-$N$ factorization for all light correlators).[2] In fact, large-$N$ factorization is not even phenomenologically desirable: the theories of quantum gravity we would most like to understand (to make a connection with our own universe), are those where a strongly interacting QFT like QCD or the Standard Model is weakly but consistently coupled to gravity. This means that matter fields in AdS should be allowed to interact at the AdS scale. In this scenario, a scalar field would instead be described by an effective Lagrangian of the form

$$\mathcal{L}_{\text{scalar}} = \frac{1}{2}\partial_\mu \phi \partial^\mu \phi + m^2 \phi^2 + g_3 \ell_{\text{AdS}}^{\frac{d-5}{2}}\phi^3 + g_4 \ell_{\text{AdS}}^{d-3}\phi^4, \tag{4}$$

for order one values of $g_{3,4}$. Such interactions require a holographic CFT where large-$N$ factorization breaks down.

A general approach for obtaining holographic CFTs without large-$N$ factorization was described in [12]. The idea is to start from a "conventional" holographic CFT that completely factorizes in the large-$N$ limit, and then turn on an exactly marginal deformation that breaks large-$N$ factorization[3,4]

$$S_{\text{CFT}} \mapsto S_{\text{CFT}} + \lambda(N) \int d^d x \, \mathbf{O}_{\text{def}}(x). \tag{5}$$

Here it is important that the deformation parameter $\lambda(N)$ depends on $N$ since distinct $N$ scalings lead to different effects in the large-$N$ limit. As shown in [19], a deformation driven by a multi-trace operator with $\lambda \sim N^0$ leads to a theory with a well-defined large-$N$ limit, but one where large-$N$ factorization is lost.

---

[1] An example of AdS/CFT where a sector of the CFT breaks large-$N$ factorization was shown to exist in type IIB string theory in [10].

[2] It is a priori not clear whether this property follows directly from (1). In $d = 2$, it does follow from (1), since all stress tensor correlators are fixed by Virasoro symmetry, but this statement does not generalize to higher dimensional CFTs where any operator can appear in the OPE of two stress tensors. A partial proof in $d > 2$ will be discussed in [11].

[3] Note that there are instances of AdS/CFT where the CFT does not completely factorize, including $\text{AdS}_5 \times S^5$ at finite string coupling, whose moduli space can contain sectors with different $N$-scalings due to coincident D3 branes, see e.g. [13].

[4] Exactly marginal multi-trace operators exist in known holographic CFTs. For example, there is such a double-trace operator in the Klebanov-Witten theory in $d = 4$ [14, 15], and marginal operators of arbitrary traces are known in symmetric orbifolds of $\mathcal{N} = 2$ minimal models in $d = 2$ [16]. In $d = 2$, double-trace deformations consisting of the product of two chiral currents are also known to be exactly marginal [17, 18].

We may be tempted to interpret the deformation (5) as a dial that continuously interpolates between CFTs whose dual effective Lagrangians start with a weakly-coupled matter sector as in (2) and "flow" to a strongly-coupled one as in (4). This would imply that the deformation can change the bulk coupling constants such that they become independent of $G_N$.[5] We know, however, that multi-trace deformations in AdS/CFT induce a change in the boundary conditions of bulk fields [20]. This suggests that the above interpretation is too naive: the breakdown of large-$N$ factorization induced by (5) is not expected to result in a local interaction on the bulk side of the AdS/CFT correspondence, at least not in conformal perturbation theory.

The aim of this paper is to shed light on this expectation. To do so, we study the bulk interpretation of the multi-trace deformations capable of breaking large-$N$ factorization. For concreteness, we will consider the simplest example of such a deformation, namely the deformation of a two-dimensional CFT with $\mathcal{N} = (2, 2)$ supersymmetry driven by the exactly marginal triple-trace operator [19]

$$\mathbf{O}_{\text{def}}(x) = \frac{1}{2\sqrt{3}} \left( :\chi \mathbf{O}\mathbf{O}: + 2 :\psi\bar{\psi}\mathbf{O}: + \text{h.c.} \right), \tag{6}$$

where $\chi$, $\psi$, and $\bar{\psi}$ are (super)descendants of a single-trace chiral primary operator $\mathbf{O}$ with $h_{\mathbf{O}} = \bar{h}_{\mathbf{O}} = 1/6$, and the deformation parameter $\lambda$ in (5) is independent of $N$. We are assuming here that the undeformed theory preserves large-$N$ factorization. As shown in [19], this deformation is exactly marginal and preserves convergence of the large-$N$ limit. Importantly, this deformation leads to a breakdown of large-$N$ factorization, as evidenced by the $\mathcal{O}(N^0)$ scaling of the OPE coefficients under the deformation [19]

$$\nabla_\lambda C_{\mathbf{O}\mathbf{O}\chi^\dagger} = \nabla_\lambda C_{\mathbf{O}\psi\bar{\psi}^\dagger} = -\frac{8\pi^4}{\Gamma(1/3)^6} + \mathcal{O}\left(N^{-1}\right). \tag{7}$$

These are the only OPE coefficients of single-trace operators affected by the deformation to leading order in the large-$N$ limit. Note that the operators featured in the OPE coefficients in (7) are special as the sum of their conformal dimensions satisfies $\Delta_{\text{total}} = 2\Delta_{\mathbf{O}} + \Delta_\chi = \Delta_{\mathbf{O}} + \Delta_\psi + \Delta_{\bar{\psi}} = 2$. This observation provides further motivation for our study since bulk Witten diagrams for cubic vertices with total conformal dimension $\Delta_{\text{total}} = d$ are divergent [21].[6] The interpretation of adding such a bulk coupling is that it leads to an anomaly, i.e. to a beta function for the corresponding multi-trace operator [22]. In this case, a nonvanishing OPE coefficient could be the result of a boundary interaction, as explicitly shown in the case of ABJM [23].

**Summary of results**

The goal of this paper is to address the fate of the bulk interactions induced by (6) by considering the four-point function of the single-trace operator $\mathbf{O}$. This four-point function depends on cross-ratios, and in the right kinematic regime can precisely probe the locality of the bulk theory. From the CFT perspective, (7) guarantees that the operator $\chi$ is exchanged in the conformal block decomposition of the four-point function. Calculating the four-point function will tell us what other operators are exchanged, and how to interpret these exchanges in the bulk theory.

---

[5]The example of AdS/CFT with a strongly coupled bulk matter sector of [10] can indeed be reached continuously. Note, however, that the strongly coupled point is separated from the factorizing point by a distance on moduli space that diverges with $N$. We will comment more on this in section 2.3.

[6]The same is true for extremal OPE coefficients, where the scaling dimension of one of the operators equals the sum of the other two.

Unlike the OPE coefficients (7), which are affected at linear order in the deformation parameter $\lambda$ at large $N$, the four-point function receives corrections starting at quadratic order in $\lambda$, and our final result reads

$$\left\langle \mathbf{O}^\dagger(0)\mathbf{O}^\dagger(x)\mathbf{O}(1)\mathbf{O}(\infty)\right\rangle_c\Big|_{\lambda^2} = \sum_{p\in\{\chi^\dagger,\mathbf{O}^2\}} \left|C_{\mathbf{O}\mathbf{O}p}\right|^2 |x|^{2(h_p-1/3)} \left|_2F_1(h_p,h_p,2h_p;x)\right|^2. \quad (8)$$

The four-point function (8) takes the form of a sum over global conformal blocks, as expected. Surprisingly, the sum over conformal blocks is finite and corresponds to the exchange of the single-trace operator $\chi^\dagger$ (as expected) along with a single double-trace operator $\mathbf{O}^2 \sim :\mathbf{O}\mathbf{O}:$. Nevertheless, this four-point function is perfectly consistent, being conformally invariant and single valued on the Euclidean plane. This last property is sensitive to the precise values of the OPE coefficients in (8) and provides a nontrivial consistency check of our results.

The four-point function (8) is of order $\mathcal{O}(N^0)$, signaling a clear breakdown of large-$N$ factorization, as anticipated from (7). This suggests that the interaction of four bulk fields dual to $\mathbf{O}$ must be strong in the sense that it is independent of $G_N$. However, the four-point function (8) cannot be interpreted as originating from a local interaction in the bulk. If it could, the correlator would exhibit a bulk-point singularity [7, 24, 25]. Here, we find no such singularity at order $\lambda^2$ since our correlator consists of a finite number of conformal blocks, each of which is regular in the bulk-point limit. Therefore, the four-point function (8) must arise from an interaction that is not localized deep in the interior of AdS. It could however be the result, for example, of a boundary term in the effective Lagrangian.

Our characterization of exactly marginal operators of arbitrary trace and their effect for various scalings at large $N$ is exhaustive. While we have not analysed higher-trace deformations explicitly, we expect similar results to those obtained here.[7] Our results suggest that exactly marginal multi-trace operators can break large-$N$ factorization but cannot induce local couplings in the bulk. Single-trace deformations, on the other hand, always preserve large-$N$ factorization.

It is important to emphasize that this analysis holds at the level conformal perturbation theory. For finite deformations (and in particular for large deformations), the analysis needs to be refined on a case by case basis. This is precisely what happens in the example of [10], which uses a single-trace transformation to break large-$N$ factorization. In that example one has to take $\lambda$ to grow with $N$ faster than naively allowed in conformal perturbation theory. We conclude that holographic CFTs dual to strongly coupled bulk matter must therefore be isolated points in the CFT landscape, or sit infinitely far apart from conventional holographic CFTs on conformal manifolds. While this conclusion is reached from explicit calculations in two-dimensions, we expect similar statements to apply to other exactly marginal multi-trace deformations that break large-$N$ factorization in higher dimensions.[8]

The paper is organized as follows. In section 2 we review the large-$N$ expansion and large-$N$ factorization of holographic CFTs. Therein we also review the effect of different kinds of exactly marginal deformations on the large-$N$ properties of these theories. In section 3 we revisit the triple-trace deformation of [19] and describe the breakdown of large-$N$ factorization deduced from the calculation of two OPE coefficients. Section 4 is devoted to the computation of four-point functions in the large-$N$ limit. We conclude in section 5 with the interpretation of our results. In appendix A we provide details on the evaluation of the integrals necessary to compute OPE coefficients and four-point functions in conformal perturbation theory.

---

[7] Marginal quadruple-trace operators were discussed in [7] and again found to affect only a finite number of conformal blocks.

[8] With the caveat that exactly marginal multi-trace deformations do not exist for all $d$, and at any given $d$ beyond two dimensions, there is a maximal possible number of traces that leads to a marginal operator. This follows from the unitarity bound.

## 2 Large-*N* CFTs and exactly marginal deformations

In this section, we review the salient features of large-*N* CFTs with a planar (i.e. 't Hooft) limit with an emphasis on large-*N* factorization. We also review some basic aspects of conformal perturbation theory and describe the interplay between the two. Finally, we describe the type of marginal operators that exist in large-*N* CFTs and the fate of large-*N* factorization under these deformations.

### 2.1 Large-*N* CFTs

Holographic CFTs are characterized by, among other properties, the existence of a dimensionless parameter $N$ that schematically controls the number of degrees of freedom of the CFT. This parameter is defined from the two-point function of the stress tensor by[9]

$$\langle T_{\mu\nu}(x^{\mu})T_{\alpha\beta}(0)\rangle = N I_{\mu\nu,\alpha\beta}(x^{\mu}), \tag{9}$$

where the function $I_{\mu\nu,\alpha\beta}(x^{\mu})$ is fixed by conformal symmetry [26]. From the holographic point of view, the AdS/CFT dictionary relates this dimensionless parameter to the scale of AdS measured in Planck units via

$$N \sim \frac{\ell_{\text{AdS}}^{d-1}}{G_N}, \tag{10}$$

where $d$ is the dimension of the CFT and we have neglected order one coefficients. Since Newton's constant controls quantum effects in gravity, such as graviton loops, the large-*N* limit corresponds to the semiclassical description of gravity in AdS.

The description of large-*N* CFTs requires taking the limit $N \to \infty$, which should be viewed as a limit in theory space. Different sectors of the theory behave quite differently under this limit. In particular, the operators of the theory split into two kinds. Operators whose scaling dimensions are fixed as $N \to \infty$ are referred to as *light* operators. An important assumption on large-*N* CFTs is that correlation functions of light operators converge in the large-*N* limit.[10] Usually, one also assumes that the number of operators of any fixed dimension converges in this limit. These assumptions exclude, for example, the *N*-fold tensor product of a given CFT.

Large-*N* CFTs also contain *heavy* operators whose scaling dimensions diverge in the large-*N* limit. These operators are interpreted as black hole microstates in the bulk theory (at least if their dimension scales linearly with $N$). We will not discuss heavy operators in this paper and will focus entirely on correlation functions of light operators.

Thus far, we have reviewed the fundamentals of large-*N* CFTs. More conditions need to be imposed for the CFT to be actually holographic, that is, to admit a bulk description in terms of semiclassical general relativity. In order to see this, note that a large-*N* gauge theory at weak coupling, e.g. $\mathcal{N} = 4$ SYM at vanishing 't Hooft coupling, complies with the above assumptions, but has a bulk dual where stringy effects are important at the AdS scale. To suppress stringy (or even just higher derivative) effects in the bulk effective field theory, one needs an extra assumption often referred to as the large gap condition [7,27]. This condition constrains the spectrum of operators with spin, and can be made precise for CFTs that satisfy large-*N* factorization, a property we now review.[11]

---

[9]This parameter is often called the central charge and sometimes denoted as $c_T$. Here, we will call it $N$, but it is not to be mistaken with the rank of a gauge group. In the case of a conformal gauge theory, the central charge is often quadratic in the rank of the gauge group, for example in $\mathcal{N} = 4$ SYM.

[10]Naively, it would appear that equation (9) does not comply with this requirement, even though the stress tensor is a light operator. This is because it has a poorly chosen normalization for large-*N* purposes. To avoid this issue, one could simply consider the normalized operator $T_{\mu\nu}/\sqrt{N}$.

[11]If the theory satisfies large-*N* factorization, then the large gap condition states that all single-trace operators of spin greater than two have a parametrically large scaling dimension [28–32].

Large-$N$ factorization is an additional condition that is often imposed for holographic CFTs. The idea is to split the light operators of the theory into two classes: single and multi-trace operators. Large-$N$ factorization means that the connected part of correlators of single-trace operators $\mathbf{O}_i$ scales with $N$ as[12]

$$\langle \mathbf{O}_1(x_1)\cdots\mathbf{O}_n(x_n)\rangle_c \sim N^{-(n-2)/2}\,. \tag{11}$$

We see that connected correlation functions are suppressed by the number of operator insertions in excess of two. In other words, the set of light single-trace operators of a holographic CFT behave as generalized free fields at infinite $N$. The leading order behavior of any correlation function is therefore obtained by Wick contracting as many pairs of operators as possible, since their two-point functions are not suppressed in $N$. For example, the $2n$-point function of single-trace operators $\mathbf{O}_i$ is given by

$$\langle \mathbf{O}_1(x_1)\cdots\mathbf{O}_{2n}(x_{2n})\rangle = \sum_{\text{perm}} \langle \mathbf{O}_{i_1}(x_{i_1})\mathbf{O}_{i_2}(x_{i_2})\rangle\cdots\langle \mathbf{O}_{i_{2n-1}}(x_{i_{2n-1}})\mathbf{O}_{i_{2n}}(x_{i_{2n}})\rangle + \mathcal{O}\left(N^{-1/2}\right),$$

where we sum over all possible permutations of the indices.

In addition to single-trace operators, crossing symmetry requires the existence of multi-trace operators $\mathbf{O}^k$ defined by

$$\langle \mathbf{O}^k(x)\mathbf{O}_1(x_1)\cdots\mathbf{O}_k(x_k)\rangle_c \sim N^0\,, \tag{12}$$

where $\mathbf{O}_1, \cdots, \mathbf{O}_k$ are k single-trace operators. It follows that, at infinite $N$, k-trace operators are just normal ordered products of singe-trace operators (along with derivatives), since

$$\langle \mathbf{O}^k(x)\mathbf{O}_1(x_1)\cdots\mathbf{O}_k(x_k)\rangle_c = \langle :\!\overbrace{\mathbf{O}_1\cdots\mathbf{O}_k}\!:(x)\mathbf{O}_1(x_1)\cdots\mathbf{O}_k(x_k)\rangle_c \sim N^0\,. \tag{13}$$

In fact, it is clear from (13) that connected correlation functions of multi-trace operators can scale like $\mathcal{O}(N^0)$ provided that the appropriate Wick contractions exist.

Thus, to leading order in the large-$N$ limit, correlation functions of single and multi-trace operators are given by sums over products of connected correlators of single-trace components. Furthermore, the leading order $N$-scaling of these correlators is determined by the terms in the sum with the maximum number of Wick contractions.

In the gravitational dual, large-$N$ factorization implies that the bulk theory of gravity is weakly coupled such that all bulk interactions, including those of matter fields, are suppressed by $1/N$, i.e. by the Planck scale. This means that all bulk fields become free in the large-$N$ limit. It is important to note, however, that a free Hilbert space for all bulk fields is a more stringent condition than necessary for a nice gravitational description of the bulk. All that is actually needed is a weakly-coupled gravitational sector, which translates into large-$N$ factorization of stress tensor correlation functions. In all prime examples of the AdS/CFT correspondence, such as gauge theories in the planar limit and symmetric product (or certain permutation) orbifolds in two dimensions, the entire light spectrum factorizes [33–36]. In these examples, all matter interactions in the bulk are suppressed by the Planck scale.

However, as discussed in [19], it is possible to relax factorization of the light spectrum such that matter interactions in the bulk are no longer suppressed and can become of order one. This is desirable, in fact, if we are interested in models of holography that more closely resemble the real world: in our universe gravitational quantum effects are weak but the matter sector (the Standard Model) is strongly interacting.

---

[12]In this paper we normalize operators such that their two-point functions are given by $\langle \mathbf{O}_i(0)\mathbf{O}_j(1)\rangle = \delta_{ij}$.

The main goal of this paper is to construct holographic CFTs that go beyond large-$N$ factorization; that is, holographic CFTs where correlation functions of some operators (dual to bulk matter fields) do not factorize at large $N$ while correlators of the stress tensor still factorize. We construct these theories by deforming "conventional" holographic CFTs, i.e. those satisfying large-$N$ factorization, with exactly marginal operators in a manner that preserves a convergent large-$N$ limit but breaks factorization in a controlled way. In the remainder of this section, we first review the different kinds of exactly marginal deformations, and then consider some aspects of conformal perturbation theory and its application in the context of large-$N$ theories.

## 2.2 Deformations of large-$N$ CFTs

Conformal field theories can admit conformal manifolds, namely operators one can deform the CFT by such that the theory remains conformal

$$S_{\text{CFT}} \mapsto S_{\text{CFT}} + \lambda \int d^d x \, \mathbf{O}_{\text{def}}(x). \tag{14}$$

For this to be the case, the scaling dimension of $\mathbf{O}_{\text{def}}$ must be equal to $d$ and must remain unchanged under the deformation. Such operators are called exactly marginal operators. Generally, it is extremely difficult to find exactly marginal operators but there are mechanisms that make this possible, most notably supersymmetry [37, 38].

For large-$N$ CFTs, exactly marginal deformations of the kind (14) need to be further specified. In particular, the $N$-scaling of the deformation parameter $\lambda$ and the trace type of the operator play a crucial role. The most general deformation takes the form

$$S_{\text{CFT}} \mapsto S_{\text{CFT}} + \lambda_k N^{\beta/2} \int d^d x \, \mathbf{O}^k(x), \tag{15}$$

where $\lambda_k$ is independent of $N$ and $\beta$ is a parameter that controls the $N$-scaling of the deformation. The operator $\mathbf{O}^k$ is an exactly marginal k-trace operator. As we will see, not all values of $\beta$ and k preserve the large-$N$ limit. Moreover, even within the values that still lead to a convergent theory, there are drastically different outcomes. We are interested in breaking large-$N$ factorization of a large-$N$ CFT through an exactly marginal deformation (15). As shown in [19], this can be achieved through certain values of $\beta$ and k.

Several examples of large-$N$ marginal deformations have been studied in the literature. The vast majority involves single-trace operators, i.e. k = 1. This is the most obvious case to consider for reasons we will explain shortly. See for example [12, 39–43] in $d = 2$ or [44] for a review on its connection to integrability in $\mathcal{N} = 4$ SYM. To the extent of our knowledge, exactly marginal multi-trace deformations with k > 2 were unknown until very recently. These deformations were discovered in symmetric product orbifolds of $\mathcal{N} = 2$ minimal models, where they can occur with arbitrary number of traces k and in many different combinations of single-trace components [16]. Multi-trace deformations with k > 2 also occur in the theories studied in [43].

## 2.3 Different scalings for the deformations

Let us now review the different values of $\beta$ and k that are allowed in (15), and their respective properties in the large-$N$ limit. Different values of the parameter $\beta$ and the number of traces k in (15) can have vastly different effects on the large-$N$ behavior of a holographic CFT. This can be seen from the large-$N$ expansion of the correlation functions of the deformed

theory. Assuming that $\lambda_k$ is small, the deformed correlation functions are given in conformal perturbation theory by

$$
\begin{aligned}
\left\langle \mathbf{O}_1(x_1)\cdots \mathbf{O}_n(x_n)\right\rangle_{\lambda_k} &= \left\langle \mathbf{O}_1(x_1)\cdots \mathbf{O}_n(x_n)\right\rangle \\
&\quad + \lambda_k N^{\beta/2} \int d^d z \left\langle \mathbf{O}^k(z)\mathbf{O}_1(x_1)\cdots \mathbf{O}_n(x_n)\right\rangle_c \\
&\quad + \lambda_k^2 N^{\beta} \int d^d z\, d^d w \left\langle \mathbf{O}^k(z)\mathbf{O}^k(w)\mathbf{O}_1(x_1)\cdots \mathbf{O}_n(x_n)\right\rangle_c \\
&\quad + \mathcal{O}(\lambda_k^3),
\end{aligned}
\tag{16}
$$

where all correlation functions on the r.h.s. are evaluated at $\lambda_k = 0$. For the deformation to be under control, the deformed theory must still have a consistent large-$N$ limit. This means that all finite quantities in the CFT must remain finite as we turn on the deformation. One way to ensure this is to set $\beta \leq 0$, since correlation functions of light operators on the r.h.s of (16) are all finite in the limit $N \to \infty$. One exception to this rule occurs for marginal single-trace deformations where $k = 1$. In this case, each $\lambda_1^n$ correction to the undeformed correlation function is suppressed by an additional factor of $N^{-n/2}$. This follows from the scaling of connected correlation functions given in (11). As a result, we can set $\beta = 1$ for single-trace deformations and still have a convergent large-$N$ limit. This is in fact the standard $N$-scaling applied to large-$N$ gauge theories in the 't Hooft limit.

Exactly marginal deformations of the form (15) give anomalous dimensions to generic operators that are not protected by symmetry. Examples of protected operators are the deformation operators themselves, which remain exactly marginal. Additionally, there can be other operators whose conformal dimensions are protected by (super)symmetry. Let $\mathbf{O}_0$ denote a non-degenerate single-trace operator with zero spin that is not protected by symmetry. The anomalous dimension $\gamma_0$ of $\mathbf{O}_0$ can be extracted from the coefficient of the logarithmic correction to the two-point function using (16)

$$
\left\langle \mathbf{O}_0(0)\mathbf{O}_0(x)\right\rangle_{\lambda_k} = \frac{1}{|x|^{2h_0+\gamma_0}} = \frac{1}{|x|^{2h_0}} - 2\gamma_0 \frac{\log |x|^2}{|x|^{2h_0}} + \mathcal{O}(\gamma_0^2),
\tag{17}
$$

where $h_0$ is the conformal weight of $\mathbf{O}_0$ in the undeformed theory. Thus, the strategy to extract anomalous dimensions is to compute the integrated correlators that appear in (16) and extract the logarithmic terms. The regularization scheme that we will employ is to cut out small balls around the location of other operators in the integrals, and throw away the divergent terms. This gives a well-defined prescription for conformal perturbation theory, and computes covariant derivatives of the correlators on the conformal manifold [45, 46].

Let us track more carefully what happens under the conformal perturbation in the large-$N$ limit. We first assume that $\mathbf{O}_0$ is a generic operator, meaning that the (potentially multi-trace) operator $\mathbf{O}^k$ does not contain $\mathbf{O}_0$ as one of its single-trace components. In this case, conformal perturbation theory gives us

$$
\begin{aligned}
\left\langle \mathbf{O}_0(x_1)\mathbf{O}_0(x_2)\right\rangle_{\lambda_k} &= \left\langle \mathbf{O}_0(x_1)\mathbf{O}_0(x_2)\right\rangle \\
&\quad + \lambda_k N^{\beta/2} \int d^d z \left\langle \mathbf{O}^k(z)\mathbf{O}_0(x_1)\mathbf{O}_0(x_2)\right\rangle_c \\
&\quad + \lambda_k^2 N^{\beta} \int d^d z\, d^d w \left\langle \mathbf{O}^k(z)\mathbf{O}^k(w)\mathbf{O}_0(x_1)\mathbf{O}_0(x_2)\right\rangle_c \\
&\quad + \mathcal{O}(\lambda_k^3).
\end{aligned}
\tag{18}
$$

In the correlators that appear on the r.h.s. of the above equation, there is no Wick contraction between $\mathbf{O}_0$ and $\mathbf{O}^k(z)$ since, by assumption, $\mathbf{O}^k(z)$ does not contain $\mathbf{O}_0$. Therefore, the integrated correlators have no $\mathcal{O}(N^0)$ contribution and $\mathbf{O}_0$ does not acquire an anomalous dimension in the large-$N$ limit. This shows that deformations with $\beta \leq 0$ cannot lead to anomalous dimensions for generic operators at large $N$. Thus, the only way to give anomalous dimensions to generic operators in the large-$N$ limit is to consider single-trace deformations which, as explained above, allow for $\beta = 1$. In the context of holography, one usually considers deformations along the conformal manifold that take a weakly-coupled CFT to a strongly-coupled point where the theory becomes holographic. In the process, most of the light operators are lifted, in particular those with spin, as required by the large gap condition. These operators correspond to stringy degrees of freedom in the bulk which become massive under the deformation. These types of deformations are always single-trace.

In this paper, we are interested in the fate of large-$N$ factorization under exactly marginal deformations. Large-$N$ factorization is completely characterized by the scaling of correlation functions in the large-$N$ limit. For simplicity, let us consider a multi-trace operator $\mathbf{O}^k$ constructed exclusively from a single-trace operator $\mathbf{O}$ such that $\mathbf{O}^k =: \mathbf{O} \cdots \mathbf{O}:$ (a similar logic applies to multi-trace operators made out of distinct constituents). After the deformation, there is a set of correlation functions whose leading order correction in the large-$N$ limit is given by (cf. the second line of (16)) [16]

$$
\begin{aligned}
N^{\beta/2} \langle \mathbf{O}^k \mathbf{O}_1 \cdots \mathbf{O}_n \rangle_c &\sim N^{\beta/2} \langle \mathbf{O} \mathbf{O}_1 \cdots \mathbf{O}_{n_1} \rangle \cdots \langle \mathbf{O} \mathbf{O}_{n_{k-1}+1} \cdots \mathbf{O}_{n_k} \rangle \\
&\sim N^{\beta/2} \prod_{j=1}^{k} N^{\frac{2-n_j-1}{2}} = N^{\frac{\beta+k-n}{2}} ,
\end{aligned}
\tag{19}
$$

where $\sum_{j=1}^{k} n_j = n$. It is important to note that (19) is the maximal possible large-$N$ scaling; not all correlation functions scale in this way, but there are always some that do. These correlation functions can be obtained by an appropriate choice of $\mathbf{O}_1, \cdots, \mathbf{O}_n$. Therefore, large-$N$ factorization is preserved if the large-$N$ scaling of (19) is the same as or slower than that of the undeformed correlator (11). It follows that large-$N$ factorization is preserved whenever $\beta \leq 2 - k$. This shows that in order to preserve large-$N$ factorization, most multi-trace deformations will have no effect in the large-$N$ limit.[13]

Note that there is an interesting range of values, namely $2 - k < \beta \leq 0$, where the deformation breaks large-$N$ factorization but preserves the large-$N$ limit. This range only appears for triple and higher-trace deformations where $k > 2$. For such exactly marginal operators, we can set $\beta = 0$ and obtain large-$N$ CFTs where a subsector of the theory breaks large-$N$ factorization. The subsector depends on the operators that make up $\mathbf{O}^k$.

To summarize, there is an important difference between single and multi-trace deformations. Single-trace deformations lead to a well-defined large-$N$ limit when $\beta \leq 1$ and always preserve large-$N$ factorization. Nevertheless, these deformations can lead to large anomalous dimensions for generic operators in the large-$N$ limit. Therefore, if we want move a from a free point on the conformal manifold where the theory has an infinite tower of higher spin currents, to a holographic point dual to semiclassical gravity, we have to use a single-trace deformation. The anomalous dimensions of higher spin currents induced by the deformations have been computed explicitly in [12, 39–43]. In some cases, these deformations can also be studied in the gravity dual close to a "tensionless" point [47–50].

---

[13]An interesting case is a deformation with $\beta = 0$ and $k = 2$. There, large-$N$ factorization would be preserved and some operators (those that make up $:\mathbf{O}^2:$) could still be affected at order $N^0$. However, if the operators that make $:\mathbf{O}^2:$ are BPS (which is how exact marginality of the deformation is assured), then nothing happens at order $N^0$, and the visible effects of the deformation are pushed to higher orders in the $1/N$ expansion.

Multi-trace deformations, on the other hand, are not capable of giving anomalous dimensions to generic operators at large $N$. However, for $k > 2$ and $\beta > 2-k$, they can break large-$N$ factorization in a subsector of the CFT. These deformations allow us to explore instances of AdS/CFT that include strong interactions in the matter sector. In the next section, we will investigate the simplest example of such a deformation that is capable of breaking large-$N$ factorization.

The analysis of this section is based on conformal perturbation theory, and studies the fate of correlation functions in the large-$N$ limit order by order in perturbation theory, demanding that each order converges. In some cases, it is possible to take scalings that grow faster with $N$, by first resumming the perturbative series, and only then taking $N \to \infty$. This is how one should view the non-factorizing theories of [10]. To reach the strongly coupled point there, one would have to make $\lambda$ scale with at least $\log(N)$. In the large-$N$ limit, perturbation theory breaks down and one has to resum the series before making $N$ large. In the process, one sector of the theory no longer factorizes at large $N$, while the rest of the CFT still factorizes.[14]

## 3 A triple-trace deformation

In the rest of the paper, we will discuss the effects of a particular multi-trace deformation in the context of two-dimensional CFTs with $\mathcal{N} = (2,2)$ supersymmetry. These multi-trace deformations are present, for example, in symmetric product orbifolds of the $\mathcal{N} = (2,2)$ minimal models.[15] These CFTs have been argued to be connected to holographic CFTs through marginal deformations that reach a holographic point somewhere on their conformal manifolds [16], as the symmetric orbifold theory contains one or more single-trace marginal operators that can be turned on and are interpreted as gauge couplings. In particular, these deformations lift the higher spin currents present at the orbifold point [12].

The theories of [16] also contain many other marginal operators which are multi-trace, and we are interested in a triple-trace deformation that breaks large-$N$ factorization. In this section, we will see that such a deformation leads to order $\mathcal{O}(N^0)$ corrections to some OPE coefficients of single-trace operators, mostly reviewing the results of [19]; in the next section we will evaluate four-point functions.

As discussed in the previous section, the simplest example of a deformation capable of breaking large-$N$ factorization is given by an exactly marginal triple-trace operator. Let us first discuss the existence of such operators. In order to ensure that the operator we deform the theory by is exactly marginal, we use supersymmetry. When the theory has at least $\mathcal{N} = (2,2)$ supersymmetry, an exactly marginal operator can be constructed from the superpartner of a chiral primary $\mathbf{O}^k$ of weight $h = \bar{h} = 1/2$ via

$$\mathbf{O}_{\text{def}} \sim G^-_{-1/2} \overline{G}^-_{-1/2} \mathbf{O}^k. \tag{20}$$

Exactly marginal operators of this type appear for arbitrary values of k in the symmetric product orbifold theories studied in [16, 43]. In this work we will focus on a particular simple deformation that was first studied in [19]. Its existence relies solely on the presence of a chiral primary $\mathbf{O}$ of weight $h_{\mathbf{O}} = \bar{h}_{\mathbf{O}} = 1/6$ in the spectrum. Using $\mathbf{O}$ we can construct the following exactly marginal triple-trace operator

$$\mathbf{O}^3 := \frac{1}{4\sqrt{3}} G^-_{-1/2} \overline{G}^-_{-1/2} :\!\mathbf{O}\mathbf{O}\mathbf{O}\!: + \text{ h.c.}, \tag{21}$$

---

[14]We thank Ofer Aharony for bringing this to our attention.

[15]The deformation we will study is present in any two-dimensional holographic CFT with a 't Hooft limit and a BPS operator of weight $h = \bar{h} = 1/6$.

where the factor of $4\sqrt{3}$ guarantees that $\mathbf{O}^3$ has unit norm. The exactly marginal deformation capable of breaking large-$N$ factorization is then given by

$$S_{\text{CFT}} \mapsto S_{\text{CFT}} + \lambda \int \mathrm{d}^2x \; \mathbf{O}^3(x). \tag{22}$$

The triple-trace operator used in (22) is one of many possible choices, since any set of single-trace chiral primaries can be used to construct it provided that their weights add up to $1/2$. Nevertheless, we expect the qualitative results of our analysis to be independent of the precise weights of the constituent operators, and it would be interesting to prove this. We also note that the deformation (22) exists in infinitely many CFTs. One family of examples is the symmetric product orbifold of the $\mathcal{N} = 2$ minimal models $A_{k+1}$ with $k = 1$ mod 3 and $k \geq 4$ (here $k$ parametrizes the value of the central charge via $c = \frac{3k}{k+2}$).

### 3.1 Single-trace OPE coefficients

In order to quantify the effects of the deformation (21), we first study its effect on OPE coefficients of single-trace operators. The effect of the deformation on single-trace OPE coefficients is determined by

$$\nabla_\lambda C_{\mathbf{O}_i \mathbf{O}_j \mathbf{O}_k} := \int \mathrm{d}^2x \left\langle \mathbf{O}^3(x)\mathbf{O}_i(\infty)\mathbf{O}_j(1)\mathbf{O}_k^\dagger(0) \right\rangle_c \Big|_{\text{reg}}, \tag{23}$$

where $C_{\mathbf{O}_i\mathbf{O}_j\mathbf{O}_k} = \left\langle \mathbf{O}_i(\infty)\mathbf{O}_j(1)\mathbf{O}_k^\dagger(0) \right\rangle$ is the OPE coefficient between $\mathbf{O}_i$, $\mathbf{O}_j$, and $\mathbf{O}_k$, while $\nabla_\lambda$ is the covariant derivative on the conformal manifold with respect to the coordinate $\lambda$. The correlation function appearing on the r.h.s of (23) is formally divergent, but there is a well-defined regularization scheme that makes the definition meaningful [45, 46]. This scheme corresponds to cutting small balls around the location of the operators to regulate the integrals.

In the undeformed theory, the OPE coefficients of single-trace operators scale as $N^{-\frac{1}{2}}$ at large $N$, as indicated in (11). A breakdown of large-$N$ factorization is signaled by a scaling of order $N^0$ of the same OPE coefficient. The only OPE coefficients of single-trace operators capable of obtaining such a correction involve the following (super)descendants of $\mathbf{O}$

$$\chi := \frac{3}{2} G_{-1/2}^- \overline{G}_{-1/2}^- \mathbf{O}, \qquad \psi := \sqrt{\frac{3}{2}} G_{-1/2}^- \mathbf{O}, \qquad \bar{\psi} := \sqrt{\frac{3}{2}} \bar{G}_{-1/2}^- \mathbf{O}. \tag{24}$$

This follows from the fact that an order $N^0$ result for (23) can only be obtained from Wick contractions of the operators $\{\mathbf{O}_i, \mathbf{O}_j, \mathbf{O}_k^\dagger\}$ with the constituents of $\mathbf{O}^3$, and that the latter can be written as

$$\mathbf{O}^3 = \frac{1}{2\sqrt{3}}\big( :\chi\mathbf{O}\mathbf{O}: + 2 :\psi\bar{\psi}\mathbf{O}: + \text{h.c.}\big). \tag{25}$$

Thus, $C_{\mathbf{O}\mathbf{O}\chi^\dagger}$ and $C_{\mathbf{O}\psi\bar{\psi}^\dagger}$ are the only OPE coefficients of single-trace operators that can obtain an order $N^0$ correction that breaks large-$N$ factorization.

In order to see that $C_{\mathbf{O}\mathbf{O}\chi^\dagger}$ indeed gets a nonzero correction at large $N$, we evaluate [19]

$$
\begin{aligned}
\left\langle \mathbf{O}^3(x)\mathbf{O}(\infty)\mathbf{O}(1)\chi(0) \right\rangle_c &= \frac{1}{2\sqrt{3}}\left\langle :\chi^\dagger\mathbf{O}^\dagger\mathbf{O}^\dagger:(x)\mathbf{O}(\infty)\mathbf{O}(1)\chi(0) \right\rangle_c + \mathcal{O}\left(N^{-1}\right) \\
&= \frac{1}{\sqrt{3}}\left\langle :\chi^\dagger\mathbf{O}^\dagger\mathbf{O}^\dagger:(x)\mathbf{O}(\infty)\mathbf{O}(1)\chi(0) \right\rangle + \mathcal{O}\left(N^{-1}\right) \\
&= \frac{1}{\sqrt{3}} \frac{1}{|x|^{8/3}|x-1|^{2/3}} + \mathcal{O}\left(N^{-1}\right),
\end{aligned}
\tag{26}
$$

where the factor of 2 between the first and second lines comes from two possible Wick contractions of the $\mathbf{O}$ operators. The only input needed to obtain (26) is the weight of $\mathbf{O}$. Since $\mathbf{O}$ is a chiral primary, and the deformation (22) preserves supersymmetry, its weight is protected on the conformal manifold. Therefore, (26) is valid under any other exactly marginal deformations that do not break large-$N$ factorization, like the standard gauge couplings built out of twist operators.

To leading order in perturbation theory we thus have

$$\nabla_\lambda C_{\mathbf{OO}\chi^\dagger}\big|_{\lambda=0} = \frac{1}{\sqrt{3}} \int \mathrm{d}^2 x \, \frac{1}{|x|^{8/3}|x-1|^{2/3}}\bigg|_{\mathrm{reg}} + \mathcal{O}(N^{-1}) \,. \tag{27}$$

This type of integral can be evaluated analytically. For reasons that will become clear later, let us define

$$\square_0(a,b) := \int \mathrm{d}^2 u \, |u|^{2a}|u-1|^{2b} \,. \tag{28}$$

The integral (28) can be computed using the methods of [19,51] or by contour integration [52,53]. Both of these methods cut out balls around the singularities to regulate the otherwise divergent integrals, with the result (see appendix A for details)

$$\square_0(a,b) = -\sin(\pi b)\frac{\Gamma(1+a)\Gamma(1+b)}{\Gamma(2+a+b)}\frac{\Gamma(-1-a-b)\Gamma(1+b)}{\Gamma(-a)} \,. \tag{29}$$

We thus find [19]

$$\nabla_\lambda C_{\mathbf{OO}\chi^\dagger}\big|_{\lambda=0} = \frac{1}{\sqrt{3}}\square_0\left(-\frac{4}{3},-\frac{1}{3}\right) + \mathcal{O}(N^{-1}) = -\frac{8\pi^4}{\Gamma(1/3)^6} + \mathcal{O}(N^{-1}) \,. \tag{30}$$

Similarly, it is not difficult to show that

$$\nabla_\lambda C_{\mathbf{O}\psi\bar\psi^\dagger}\big|_{\lambda=0} = \frac{1}{\sqrt{3}} \int \mathrm{d}^2 x \, \frac{x(\bar{x}-1)}{|x-1|^{8/3}|x|^{8/3}} + \mathcal{O}(N^{-1}) = -\frac{8\pi^4}{\Gamma(1/3)^6} + \mathcal{O}(N^{-1}) \,. \tag{31}$$

The corrections to the OPE coefficients (30) and (31) are the same, as they must be related by supersymmetry. Note that this result is a genuine change in the OPE coefficient and not a consequence of mixing between different operators, the effects of which only appear at $\mathcal{O}(N^{-1})$ [54,55].

We have thus found a pair of OPE coefficients that, in the infinite-$N$ limit, vanish at $\lambda = 0$ but receive order $N^0$ corrections when we turn on the deformation. All other OPE coefficients of primary single-trace operators are $N$-suppressed before turning on the deformation and remain suppressed under it. Thus, the deformation breaks large-$N$ factorization, but it does so in a controlled way: only the sector of the CFT containing operators made from $\mathbf{O}$ can obtain finite couplings at infinite $N$.

It is important to note that there are certain OPE coefficients of multi-trace operators that are already finite at $\lambda = 0$ in the large-$N$ limit. One example is $C_{\mathbf{OOO}^2} = \sqrt{2}$, where $\mathbf{O}^2 := \frac{1}{\sqrt{2}} :\mathbf{OO}:$ is the simplest double-trace operator constructed from $\mathbf{O}$. Such OPE coefficients can get finite corrections under the deformation, as we will see explicitly in the next section.

## 4 Four-point functions

In the previous section, we considered a triple-trace deformation of two-dimensional CFTs that breaks large-$N$ factorization in a subsector of the theory. The gravitational interpretation of

this result is subtle. One may be tempted to interpret the order one OPE coefficients as a sign of local bulk interactions, consistent with (4). However, a cubic interaction for the fields dual to $\mathbf{O}$, $\mathbf{O}$, and $\chi$ (or $\mathbf{O}$, $\psi$, and $\bar{\psi}$) would lead to an IR divergence in the corresponding Witten diagram, which stems from the fact that the scaling dimensions of the operators featured in the interaction add up to $d = 2$ [21]. Hence, finite bulk cubic interactions cannot account for the breakdown of large-$N$ factorization. Instead, as described in the introduction, a bulk interaction of this kind is expected to lead to an anomaly [22], see also [56] for a recent discussion.

Following [20] and related work, we expect that the OPE coefficients described in the previous section are the result of an interaction that is localized on the boundary, rendering the breakdown of large-$N$ factorization a boundary effect, at least at the level of three-point functions. Although the corrections to the OPE coefficients cannot arise from a local three-point interaction in the bulk, it is still possible that it leads to higher-order local bulk couplings. Investigating whether this scenario is realized in our setup is the goal of this section.

Local bulk interactions can be probed by certain singularities in the dynamical correlators of the dual CFT. These so-called bulk-point singularities only arise when the boundary operators are aligned such that they allow for classical scattering in the bulk interior [7, 24, 25]. If these singularities are not present, there are no classical scattering processes and hence no local interactions in the low energy effective bulk theory. Therefore, in order to understand the bulk interpretation of the finite OPE coefficients induced by multi-trace deformations, we need dynamical data of the deformed CFT. The simplest correlation functions that contain dynamical information are four-point functions, which are the subject of this section.

Before we consider four-point functions in the presence of the deformation (22), we need to discuss a generalization of the integral $\square_0(a, b)$. Let us define[16]

$$\square(a, b, c; x, \bar{x}) := \int \mathrm{d}^2 u |u|^{2a} |u - 1|^{2b} |u - x|^{2c}. \tag{32}$$

This integral is formally divergent but can be regulated by cutting out balls around the singularities. We can evaluate this integral by the contour integration method of [52, 53], which automatically takes care of regularization. In this case, (32) can be written in terms of two one-dimensional integrals such that [52, 53] (see appendix A for details)

$$\square(a, b, c; x, \bar{x}) = \frac{\sin(\pi b)\sin(\pi(a + b + c))}{\sin(\pi(a + c))} |I_1(a, b, c; x)|^2 + \frac{\sin(\pi a)\sin(\pi c)}{\sin(\pi(a + c))} |I_2(a, b, c; x)|^2, \tag{33}$$

where $I_1(a, b, c; x)$ and $I_2(a, b, c; x)$ are given by

$$I_1(a, b, c; x) := \int_1^\infty \mathrm{d}u\, u^a (u - 1)^b (u - x)^c, \quad I_2(a, b, c; x) := \int_0^x \mathrm{d}u\, u^a (1 - u)^b (x - u)^c. \tag{34}$$

These integrals can be evaluated explicitly and are given in terms of hypergeometric functions

$$
\begin{aligned}
I_1(a, b, c; x) &= B(-1 - a - b - c, b + 1)\,_2F_1(-c, -a - b - c - 1, -a - c; x), \\
I_2(a, b, c; x) &= x^{a+c+1} B(a + 1, c + 1)\,_2F_1(-b, a + 1, a + c + 2; x),
\end{aligned}
\tag{35}
$$

where $B(x, y) = \frac{\Gamma(x)\Gamma(y)}{\Gamma(x+y)}$ is the beta function.

Let us come back to the correlation functions of the deformed theory. One four-point function that receives corrections at order $N^0$ is $\langle \mathbf{O}^\dagger(0)\mathbf{O}^\dagger(x)\mathbf{O}(1)\mathbf{O}(\infty)\rangle_c$. The leading correction to this four-point function is found at second order in $\lambda$ and is determined entirely by Wick

---

[16]Note that $\square_0(a, b) = \square(a, b, 0; x, \bar{x})$ is the integral (28) used in the evaluation of $C_{\mathbf{OO}\chi^\dagger}$.

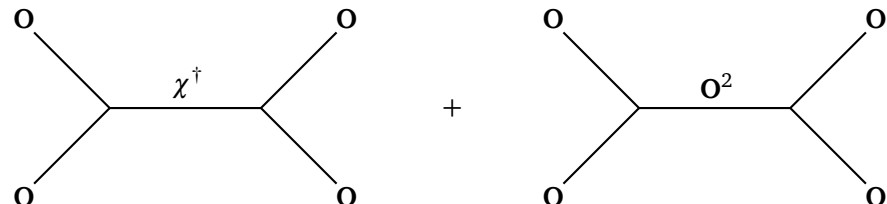

Figure 1: The exchange diagrams contributing to $\langle \mathbf{O}^\dagger(0)\mathbf{O}^\dagger(x)\mathbf{O}(1)\mathbf{O}(\infty)\rangle_c\big|_{\lambda^2}$.

contractions of the operators making up the deformation (22). Omitting $\mathcal{O}(N^{-1})$ corrections, we obtain the order $\lambda^2$ contribution

$$
\begin{aligned}
\langle \mathbf{O}^\dagger(0)\mathbf{O}^\dagger(x)\mathbf{O}(1)\mathbf{O}(\infty)\rangle_c\big|_{\lambda^2} &= \frac{\lambda^2}{3}\int d^2w \int d^2z \,\langle :\chi\mathbf{OO}:(w):\chi^\dagger\mathbf{O}^\dagger\mathbf{O}^\dagger:(z)\,\mathbf{O}^\dagger(0)\mathbf{O}^\dagger(x)\mathbf{O}(1)\mathbf{O}(\infty)\rangle \\
&= \frac{\lambda^2}{3}\int d^2w \int d^2z \,\frac{1}{|w-z|^{8/3}|w|^{2/3}|w-x|^{2/3}|z-1|^{2/3}} \\
&= \frac{\lambda^2}{3}\int d^2w'\,\frac{1}{|w'|^{2/3}|w'-1|^{4/3}|w'-x|^{2/3}}\int d^2z'\,\frac{1}{|z'|^{8/3}|z'-1|^{2/3}} \\
&= \frac{\lambda^2}{3}\Box\!\left(-\frac{1}{3},-\frac{2}{3},-\frac{1}{3};x,\bar{x}\right)\Box_0\!\left(-\frac{4}{3},-\frac{1}{3}\right),
\end{aligned}
$$
(36)

where in the third line we changed variables $(z,w)=(w'+z'(w'-1),w')$ in order to get an expression that depends explicitly on the regulated integrals $\Box(a,b,c;x,\bar{x})$ and $\Box_0(a,b)$. Using (29) and (33) we then obtain

$$
\langle \mathbf{O}^\dagger(0)\mathbf{O}^\dagger(x)\mathbf{O}(1)\mathbf{O}(\infty)\rangle_c\big|_{\lambda^2} = \left|C_{\mathbf{OO}\chi^\dagger}\,x^{1/3}\,{}_2F_1\!\left(\frac{2}{3},\frac{2}{3},\frac{4}{3};x\right)\right|^2 - 3\pi^2\lambda^2\left|{}_2F_1\!\left(\frac{1}{3},\frac{1}{3},\frac{2}{3};x\right)\right|^2,
$$
(37)

where $C_{\mathbf{OO}\chi^\dagger}$ is the OPE coefficient computed in the previous section. Recall that $C_{\mathbf{OO}\chi^\dagger}\sim\lambda$, so both terms in (37) are $\mathcal{O}(\lambda^2)$.

We conclude that the four-point function (37) is the sum of two conformal blocks that can be compactly written as

$$
\langle \mathbf{O}^\dagger(0)\mathbf{O}^\dagger(x)\mathbf{O}(1)\mathbf{O}(\infty)\rangle_c\big|_{\lambda^2} = \sum_{p\in\{\chi^\dagger,\mathbf{O}^2\}}\left|C_{\mathbf{OO}p}\right|^2|x|^{2(h_p-1/3)}\left|{}_2F_1(h_p,h_p,2h_p;x)\right|^2.
$$
(38)

The first block in (38) is the exchange of the $\chi^\dagger$ operator in the $s$-channel, while the second block is interpreted as the exchange of the double-trace operator $\mathbf{O}^2 = \frac{1}{\sqrt{2}}:\mathbf{OO}:$. In particular, (36) encodes the change in the OPE coefficient $C_{\mathbf{OOO}^2}$ after the deformation such that

$$
C_{\mathbf{OOO}^2} = \sqrt{2} - \frac{3\pi^2\lambda^2}{2\sqrt{2}} + \mathcal{O}(\lambda^4).
$$
(39)

The two exchange diagrams are depicted in figure 1.

Although (38) only receives contributions from two conformal blocks at leading order in the large-$N$ limit, this is a perfectly consistent four-point function, being conformally invariant and single valued. Indeed, it is not difficult to verify that the four-point function is invariant under conformal transformations as can be seen from the analogous computations

$$
\langle \mathbf{O}(0)\mathbf{O}^\dagger(x)\mathbf{O}^\dagger(1)\mathbf{O}(\infty)\rangle_c\big|_{\lambda^2} = \sum_{p\in\{\chi^\dagger,\mathbf{O}^2\}}\left|C_{\mathbf{OO}p}\right|^2|1-x|^{2(h_p-1/3)}\left|{}_2F_1(h_p,h_p,2h_p;1-x)\right|^2,
$$

and

$$\left\langle \mathbf{O}(0)\mathbf{O}^\dagger(x)\mathbf{O}(1)\mathbf{O}^\dagger(\infty)\right\rangle_c\Big|_{\lambda^2} = |x|^{-2/3}\sum_{p\in\{\chi^\dagger,\mathbf{O}^2\}}\left|C_{\mathbf{O}\mathbf{O}p}\right|^2|x|^{-2(h_p-1/3)}\left|{}_2F_1(h_p,h_p,2h_p;x^{-1})\right|^2.$$

Note that both of these expressions, together with (38), are valid to leading order in the large-$N$ expansion and to quadratic order in the deformation parameter $\lambda$.

The four-point function (38) is also single valued. This is harder to establish than conformal invariance, and it cannot be realized as a finite sum over conformal blocks for general scaling dimensions. In our case, single valuedness relies on the precise ratio between the prefactors of the hypergeometric functions. Single valuedness around $x = 0$ is not difficult to see from the explicit expression (26). In order to see that the four-point function is also single valued around $x = 1$, we need to use the following identity

$$\begin{aligned}{}_2F_1(a,b;c;z) &= \frac{\Gamma(c)\Gamma(a+b-c)}{\Gamma(a)\Gamma(b)}(1-z)^{-a-b+c}{}_2F_1(c-a,c-b;-a-b+c+1;1-z)\\&\quad + \frac{\Gamma(c)\Gamma(-a-b+c)}{\Gamma(c-a)\Gamma(c-b)}{}_2F_1(a,b;a+b-c+1;1-z).\end{aligned} \tag{40}$$

Using (40), we find that the $I_1(a,b,c;x)$ and $I_2(a,b,c;x)$ integrals satisfy

$$\begin{aligned}I_1(a,b,c;x) &= -\frac{\sin(\pi c)}{\sin(\pi(b+c))}I_2(b,a,c;1-x) + \frac{\sin(\pi a)}{\sin(\pi(b+c))}I_1(b,a,c;1-x),\\I_2(a,b,c;x) &= -\frac{\sin(\pi b)}{\sin(\pi(b+c))}I_2(b,a,c;1-x) - \frac{\sin(\pi(a+b+c))}{\sin(\pi(b+c))}I_1(b,a,c;1-x).\end{aligned} \tag{41}$$

Thus, $\square(a,b,c,x,\bar{x})$ can be equivalently written as

$$\square(a,b,c,x,\bar{x}) = b_1|I_1(b,a,c,1-x)|^2 + b_2|I_2(b,a,c,1-x)|^2, \tag{42}$$

where the coefficients $b_{1,2}$ are equal to ratios of sine functions that ensure a trivial monodromy around $x = 1$. Therefore, the integrals featured in the evaluation of the deformed four-point function have single valuedness built in.

Our result is also compatible with crossing symmetry. In the cross-channel where the operators $\mathbf{O}$ and $\mathbf{O}^\dagger$ fuse, the two conformal blocks we have obtained in the original channel will lead to anomalous dimensions and changes in the OPE coefficients of the double-trace operators $:\mathbf{O}\mathbf{O}^\dagger:$. This is in agreement with crossing symmetry and can be checked operator by operator.

The features described above are not unique to the four-point function (37). Another four-point function that gets corrected at order $\mathcal{O}(N^0)$ is $\langle\chi^\dagger(0)\mathbf{O}^\dagger(x)\chi(1)\mathbf{O}(\infty)\rangle_c$. The computation of this correlator is analogous to the previous one, and it also results in a sum over two conformal blocks

$$\left\langle\chi^\dagger(0)\mathbf{O}^\dagger(x)\chi(1)\mathbf{O}(\infty)\right\rangle_c\Big|_{\lambda^2} = \sum_{p\in\{\mathbf{O}^\dagger,:\chi\mathbf{O}:\}}\left|C_{\chi\mathbf{O}p}\right|^2|x|^{2(h_p-5/6)}\left|{}_2F_1\left(h_p-\frac{1}{2},h_p+\frac{1}{2},2h_p;x\right)\right|^2.$$

This four-point function features an $\mathbf{O}^\dagger$ exchange with the expected OPE coefficient $C_{\chi\mathbf{O}\mathbf{O}^\dagger} = C_{\mathbf{O}\mathbf{O}\chi^\dagger}$. Moreover, there is a conformal block associated with a $:\chi\mathbf{O}:$ exchange with OPE coefficient

$$C_{\chi\mathbf{O}:\chi\mathbf{O}:} = 1 - \frac{3\pi^2\lambda^2}{8} + \mathcal{O}(\lambda^4). \tag{43}$$

This double-trace OPE coefficient gets adjusted in the right way to make the four-point function single valued. A simple check shows that the four-point function (4) is also invariant under conformal transformations. The two exchanges contributing to $\langle\chi^\dagger(0)\mathbf{O}^\dagger(x)\chi(1)\mathbf{O}(\infty)\rangle_c\big|_{\lambda^2}$ are shown in figure 2.

This concludes our computations of the four-point functions. The interpretation of these results and a discussion of open questions are considered in the next section.

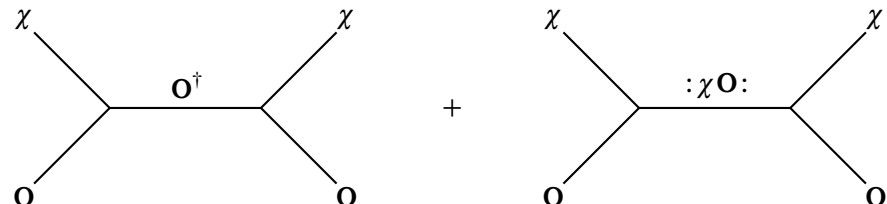

Figure 2: The exchange diagrams contributing to $\left\langle \chi^\dagger(0)\mathbf{O}^\dagger(x)\chi(1)\mathbf{O}(\infty)\right\rangle_c\big|_{\lambda^2}$.

## 5  Discussion

We have shown that an exactly marginal triple-trace deformation (the deformation (22)) leads to well-defined nonvanishing connected four-point functions at infinite $N$. This is a clear breakdown of large-$N$ factorization, which states that the only nonvanishing part of a four-point function at large $N$ is its disconnected component. Interestingly, the four-point functions we have computed only receive contributions from two conformal blocks: the *minimal* exchange of an operator consistent with the OPE coefficients computed in section 3 and a single other operator. Our result satisfies the axiomatic rules of CFT, in particular single valuedness of the four-point function on the Euclidean plane. In this sense, the result we have found is the simplest way to preserve single valuedness given the exchange of the operator found in section 3. We conclude with the interpretation of our results and open questions.

**No bulk-point singularity**

Our results immediately suggest that the triple-trace deformation does not induce a local interaction in the bulk, at least not in perturbation theory. Such a local bulk interaction would correspond to a sum over an infinite number of conformal blocks. Indeed, a finite number of blocks in the Euclidean four-point function cannot lead to the bulk-point singularities characteristic of a local quartic interaction in the bulk [7, 24, 25]. This follows from the fact these singularities arise from the non-holomorphic behavior in the Lorentzian continuation of the Euclidean four-point function. Such a behavior is not possible when the Euclidean correlator features only a finite number of products of holomorphic and antiholomorphic functions as in (37).

   The expectation for the nature of the bulk theory is that it includes a boundary interaction for the bulk matter fields of the form

$$S_{\text{bulk}} \to S_{\text{bulk}} + \int_{\partial\text{AdS}} \mathrm{d}t_{\text{E}}\, \mathrm{d}x \; \phi^3(z=0, t_{\text{E}}, x), \qquad (44)$$

where $\partial\text{AdS}$ denotes the boundary of AdS at $z = 0$. This boundary interaction implements the change of boundary conditions of the bulk fields induced by the deformation (22).

   The situation we find in the bulk is reminiscent of boundary conformal field theory (BCFT), where the boundary degrees of freedom can be strongly coupled, while the bulk degrees of freedom are not necessarily so. This scenario can arise, for example, in a free theory with conformal boundary conditions, where one turns on a boundary marginal operator that drastically changes the boundary data. In such a setup, the bulk spectrum and OPE are completely unchanged, and thus remain free. However, if one only studies the boundary data, it may be very difficult to see whether the bulk sector is free or not. In our context, the bulk-point limit gives us a tractable observable that encodes the relevant physics, which has no natural counterpart in BCFT.

### The bulk point at finite coupling

An interesting question is to understand the fate of the theory at finite $\lambda$. There, we expect that infinitely many conformal blocks contribute to the four-point function and the fate of the bulk-point singularity is more obscure. Note that the operators which contribute in the four-point function are higher and higher trace operators, rather than double-trace operators $:\mathbf{O}(\partial_\mu \partial^\mu)^n \mathbf{O}:$ with higher and higher $n$. It would be interesting to see if this makes a difference in the bulk-point limit, as it seems possible to us for the singularity to emerge again at finite $\lambda$ (but still large $N$).

### More general multi-trace deformations

A natural question to ask is whether the results obtained here are a generic feature of exactly marginal multi-trace deformations that break large-$N$ factorization, or special features of the triple-trace deformation considered herein. At least in the case of two dimensions, conformal perturbation theory suggests this is a generic feature of multi-trace deformations. Let us first consider the case where the four-point function is obtained from Wick contractions of scalar operators. In this case, the leading $\mathcal{O}(N^0)$ contribution to the four-point function of single-trace operators can be represented in terms of products of the integrals $\square(a, b, c; x, \bar{x})$. This follows from the fact that exact marginality requires the deforming operator to be a supersymmetric descendant of a chiral primary so that the leading $\mathcal{O}(N^0)$ correction to the four-point function is of $\mathcal{O}(\lambda^2)$. As a result, the four-point function is given by a product of $\square(a, b, c; x, \bar{x})$ which can be written as a finite sum over conformal blocks.

For Wick contractions between operators with spin, the only change in the argument is in the form of the integrand, which is no longer a real function. While we have not evaluated the general form of these integrals explicitly, they satisfy a factorization property similar to the one found in the evaluation of $\square_0(a, b)$ and $\square(a, b, c; x, \bar{x})$ (see appendix A). For this reason, we expect these integrals to yield similar results such that the four-point function is given by a finite sum over conformal blocks.[17] Consequently, we do not expect to see the emergence of a bulk-point singularity in the four-point function of theories deformed by higher trace deformations, at least in perturbation theory.

It is worthwhile to mention that the case of a quadruple-trace deformation was discussed in [7], and there again the CFT data was modified in such a way that a single OPE coefficient is affected to first order in conformal perturbation theory. This parallels nicely with the results obtained in this paper.

### Conformal manifolds and infinite distances

If even at finite $\lambda$ there is no way to interpret our deformed CFTs as being dual to bulk matter sectors with local interactions, we are faced with a no-go result: it is not possible to obtain a CFT dual to a strongly coupled bulk matter from a CFT with a 't Hooft like expansion in perturbation theory. Perturbatively, single-trace deformations always preserve the 't Hooft limit and large-$N$ factorization, while we have shown that multi-trace deformations appear to not produce local bulk interactions.

To be clear, this does not mean that such holographic CFTs do not exist. Concretely, one can imagine a couple of avenues that evade this conclusion. For example, one could imagine changing the order of limits. This entails deforming the theories while keeping $N$ finite, resumming the perturbative expansion, and only then taking the large-$N$ limit. Moreover, one

---

[17]As an explicit example of this, consider integrals determining the correction to the OPE coefficients $C_{\mathbf{O}\mathbf{O}\chi^\dagger}$ and $C_{\mathbf{O}\psi\tilde{\psi}^\dagger}$ in (27) and (31). The first integrand is real (the result of Wick contractions between scalar operators) while the second integrand is complex (obtained from Wick contractions of operators with spin). Both integrals are related to $\square_0\left(-\frac{4}{3}, -\frac{1}{3}\right)$ and yield the same answer.

could obtain a strongly coupled bulk matter sector by non-perturbative procedures, such as performing an $S$-transformation on moduli [10]. Both of these strategies illustrate that on a conformal manifold, CFTs dual to strongly coupled bulk matter sectors are separated by a distance that diverges with $N$ from CFTs satisfying large-$N$ factorization. Therefore, these phenomenologically interesting CFTs are in some sense isolated.

Another interesting possibility is to deform a standard holographic CFT with a relevant operator and flow to the IR. One could also add probe branes to the bulk on which a matter theory is strongly coupled. The addition of branes suggests that such theories are "isolated" in the QFT landscape sense, and are separated from factorizing points on the moduli space by distances that diverge with $N$.

### Naturalness from string compactifications

An interesting question related to this work is whether matter in AdS that is coupled at the Planck scale is more natural in a landscape sense, and whether having strongly coupled matter requires fine-tuning. An argument from string compactifications indicates that this could be the case. In general, the low energy effective action of string compactifications (with fluxes) takes the schematic form

$$S_{\text{EFT}} = -\frac{1}{16\pi G_N} \int \mathrm{d}^D x \sqrt{g}(R + V(\phi)), \tag{45}$$

where $D$ is the number of large (i.e. not string or Planck scale) dimensions, and $V(\phi)$ is a potential for scalar fields which, importantly, should be seen as including their kinetic terms. If the potential contains a constant term, the theory admits an AdS vacuum. For the vacuum to have a large AdS scale, it requires the length scale of the potential to be parametrically larger than the Planck scale (in the case of $\text{AdS}_5 \times S^5$, this is achieved by putting $N$ units of flux). This parametric separation requires some sort of fine-tuning, and can be understood as the cosmological constant problem. Of course, the problem is "resolved" by the large-$N$ condition on the CFT side.

If one further assumes that there are no other length scales in the potential, then one is naturally led to a gravitational theory where the matter interacts at the Planck scale, and where the dual CFT factorizes at large $N$. This can be seen by rescaling the scalar fields such that their kinetic terms are canonically normalized, which induces Planckian factors in the interactions. One is thus led to wonder whether strongly coupled matter requires further fine-tuning, beyond solving the cosmological constant problem; or said differently, it reinstates the cosmological constant problem that large $N$ was previously solving. It would be very interesting to understand this question better.

## Acknowledgments

We thank Agnese Bissi, Alejandra Castro, Ofer Aharony, Nadav Drukker, Christoph Keller, Shota Komatsu, Vasilis Niarchos, Kyriakos Papadodimas, Eric Perlmutter, Allic Sivaramakrishnan, Alessandro Tomasiello, and Sasha Zhiboedov for helpful discussions.

**Funding information** The work of LA was supported in part by the Dutch Research Council (NWO) through the Scanning New Horizons programme (16SNH02). The work of SB is supported by the Delta ITP consortium, a program of the Netherlands Organisation for Scientific Research (NWO) that is funded by the Dutch Ministry of Education, Culture and Science (OCW). AB and SB would like to thank the Isaac Newton Institute for Mathematical Sciences,

Cambridge, for support and hospitality during the programme "Black holes: bridges between number theory and holographic quantum information" where work on this paper was undertaken. This work was supported by EPSRC grant no EP/R014604/1.

## A  Evaluation of integrals

In this appendix we explain some of the steps necessary to evaluate the integral (32), which we reproduce here for convenience

$$\Box(a, b, c; x, \bar{x}) := \int d^2u |u|^{2a} |u-1|^{2b} |u-x|^{2c} . \tag{A.1}$$

First, let us compute the simpler integral $\Box_0(a, b) = \Box(a, b, 0; x, \bar{x})$ using contour integrals, as is done in [52, 53]. The integral (A.1) can then be evaluated using the same methods. As shown in [12], $\Box_0(a, b)$ can also be evaluated using polar coordinates by introducing a cutoff near the location of the singularity at $u = 1$ or $u = 0$, and then removing the divergent term proportional to an inverse power of the cutoff. This method was recently generalized in [51].

The integral $\Box_0(a, b)$ is given by

$$\Box_0(a, b) := \int d^2u |u|^{2a} |u-1|^{2b} . \tag{A.2}$$

Here we consider the case where $a, b < 0$. The first step in the evaluation of the integral is to change coordinates to $u = u_1 + iu_2$, which results in

$$\Box_0(a, b) = i \int_{-\infty}^{\infty} du_1 \int_{-\infty}^{\infty} du_2 \left(u_1^2 + u_2^2\right)^a \left((u_1 - 1)^2 + u_2^2\right)^b . \tag{A.3}$$

Next, we analytically continue $u_2 \longmapsto ie^{-2i\varepsilon}u_2$, with $\varepsilon > 0$, so that

$$\Box_0(a, b) = i \int_{-\infty}^{\infty} du_1 \int_{-\infty}^{\infty} du_2 \left(u_1^2 + u_2^2 e^{-4i\varepsilon}\right)^a \left((u_1 - 1)^2 + u_2^2 e^{-4i\varepsilon}\right)^b . \tag{A.4}$$

We then expand around $\varepsilon = 0$ and notice that, up to $\mathcal{O}(\varepsilon)$ corrections, the integrals factorize once we consider lightcone coordinates $u_\pm = u_1 \pm u_2$

$$\begin{aligned}
\Box_0(a, b) = \frac{i}{2} \int_{-\infty}^{\infty} &du_+ (u_+ - i\varepsilon(u_+ - u_-))^a (u_+ - 1 - i\varepsilon(u_+ - u_-))^b \\
\times \int_{-\infty}^{\infty} &du_- (u_- + i\varepsilon(u_+ - u_-))^a (u_- - 1 + i\varepsilon(u_+ - u_-))^b .
\end{aligned} \tag{A.5}$$

Due to the analytic continuation, the integrals in (A.5) should be seen as contour integrals on the complex plane with poles or branch cuts emanating from 0 and 1. The terms multiplied by $\varepsilon$ give a prescription as to how to go around the poles. There are the following three possibilities:

- **$u_+ < 0$.** The contour of $u_-$ is below the real axis at $u_- = 0, 1$.

- **$0 < u_+ < 1$.** The contour of $u_-$ is above the real axis at $u_- = 0$, and below it at $u_- = 1$.

- **$u_+ > 1$.** The contour of $u_-$ is above the real axis at $u_- = 0, 1$.

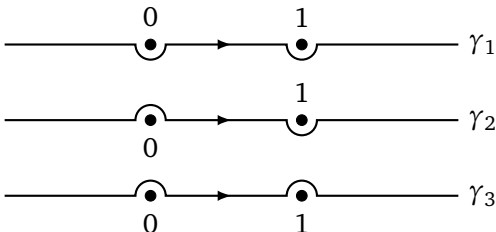

Figure 3: The contours $\gamma_{1,2,3}$ in the complex $u_-$ plane.

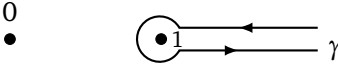

Figure 4: The deformation of the $\gamma_2$ contour in the complex $u_-$ plane.

Therefore, the integral over $u_+$ splits into three qualitatively different parts, each with their own contour (shown in figure 3)

$$\int_{-\infty}^{0} du_+ \cdots \int_{\gamma_1} du_- \cdots + \int_{0}^{1} du_+ \cdots \int_{\gamma_2} du_- \cdots + \int_{1}^{\infty} du_+ \cdots \int_{\gamma_3} du_- \cdots . \tag{A.6}$$

The contours $\gamma_1$ and $\gamma_3$ can be deformed to infinity, where the integrand vanishes, while the $\gamma_2$ contour is deformed to the contour $\gamma$ shown in figure 4. The deformed contour can be parametrized as $\gamma = \gamma_+ \cup \gamma_0 \cup \gamma_-$ where $\gamma_0$ and $\gamma_\pm$ are given by

$$\begin{aligned}
\gamma_+(t) &= i\varepsilon - te^{i\varepsilon}, & t &\in \mathbb{R}_{\leq -1}, \\
\gamma_0(\varphi) &= 1 + |\varepsilon|e^{i\varphi}, & -\frac{\pi}{2} &\leq \varphi \leq \frac{\pi}{2}, \\
\gamma_-(t) &= -i\varepsilon + te^{2\pi i - i\varepsilon}, & t &\in \mathbb{R}_{\geq 1}.
\end{aligned} \tag{A.7}$$

In the limit $\varepsilon \to 0$, the contribution of $\gamma_0$ to the integral is a pure divergence, while the contributions from $\gamma_\pm$ are finite, and combine to

$$\lim_{\varepsilon \to 0} \int_{\gamma_+ \cup \gamma_-} du_- (u_-)^a (u_- - 1)^b = -\sin(\pi b) \int_{1}^{\infty} dt \; t^a (t-1)^b . \tag{A.8}$$

The full integral is regulated by subtracting the purely divergent contribution coming from $\gamma_0$. This regularization scheme amounts to minimal subtraction and is compatible with the one used previously in [12]. After combining all of the above and changing coordinates to $x = 1/t$ we thus find

$$\Box_0(a, b) = -\int_{0}^{1} du_+ u_+^a (u_+ - 1)^b \sin(\pi b) \int_{0}^{1} dx \; x^{-2-a-b}(x-1)^b . \tag{A.9}$$

This integral evaluates straightforwardly to

$$\Box_0(a, b) = -\sin(\pi b) \frac{\Gamma(1+a)\Gamma(1+b)}{\Gamma(2+a+b)} \frac{\Gamma(-1-a-b)\Gamma(1+b)}{\Gamma(-a)} . \tag{A.10}$$

Let us now turn to the integral $\Box(a, b, c; x, \bar{x})$ defined in (A.1). Using the same analytic continuation used to evaluate $\Box_0(a, b)$ above, and similar steps and manipulations, one finds that there are two contours that contribute nontrivially to $\Box(a, b, c; x, \bar{x})$. In analogy with the evaluation of $\Box_0(a, b)$, we can deform the contours to enclose one of the poles in the integral,

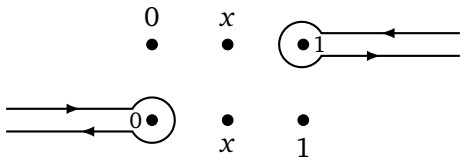

Figure 5: Deformations of the different contours that contribute to $\square(a,b,c;x,\bar{x})$.

see figure 5. By carefully parametrizing these contours, one then obtains (33), which we reproduce here for completeness [53]

$$\square(a,b,c;x,\bar{x}) = \frac{\sin(\pi b)\sin(\pi(a+b+c))}{\sin(\pi(a+c))}|I_1(a,b,c;x)|^2 + \frac{\sin(\pi a)\sin(\pi c)}{\sin(\pi(a+c))}|I_2(a,b,c;x)|^2,$$

where $I_1(a,b,c;x)$ and $I_2(a,b,c;x)$ are given in terms of $B(x,y) = \frac{\Gamma(x)\Gamma(y)}{\Gamma(x+y)}$ by

$$\begin{aligned}
I_1(a,b,c;x) &= B(-1-a-b-c,b+1)\,_2F_1(-c,-a-b-c-1,-a-c;x),\\
I_2(a,b,c;x) &= x^{a+c+1}B(a+1,c+1)\,_2F_1(-b,a+1,a+c+2;x).
\end{aligned} \tag{A.11}$$

In particular, note that the regularization scheme used in the evaluation of this integral is equivalent to minimal subtraction, namely to the removal of the purely divergent contributions to the integral.

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
