# Peer review of "Searching for strongly coupled AdS matter with multi-trace deformations"

_SciPost Physics, doi:SciPost Phys. 19, 139 (2025)_

## Round 3 · Referee Report · Anonymous (Referee 1) · 2025-8-15

Strengths

1) A clear analysis of the N-scaling of multi-trace deformations, and what needs to be done to modify the standard scalings.

2) An explicit analysis of a triple-trace deformation and how it changes 3-point and 4-point functions, in a way that is not consistent with its interpretation as a modification of bulk couplings in a holographic dual.

Weaknesses

1) Some issues are not discussed clearly, as detailed in the report below.

Report

The paper discusses whether strongly coupled theories on anti-de Sitter space (in the context of holography) may be achieved by deforming a weakly coupled theory (of the type that typically arises as the dual to large $N$ gauge theories). The authors give general arguments that in order for a modification of the $1/N$ expansion (related to perturbation theory in the bulk) to arise in a perturbative expansion, one has to turn on multi-trace operators with non-standard $N$-scalings of their coefficients, but they then show that such deformations do not modify the bulk action so that they are related to boundary interactions. While the conclusions of the paper are not surprising, it is still useful to have an organized analysis of this issue supported by explicit computations, so I recommend the publication of this paper.

There are a few issues which I believe should be clarified before the paper is published.

The main issue is the question of whether one would expect multi-trace deformations, even when they are large so that they affect the $1/N$ expansion, to modify the bulk physics. To me it seems clear that they cannot do so, certainly in a perturbation expansion, since we know that the leading order deformation can be realized as a modification of the boundary conditions (independently of the $N$-scaling of the coupling), as mentioned in the introduction. It thus seems strange that the authors write in section 4 that ``One possibility" is that the deformation involves changes on the boundary, even though we know that perturbatively this reproduces the deformation, so it is the obvious interpretation and not just a possibility. I think the authors should stress more the basis for the expectation that such deformations would arise through boundary deformations, and the fact that a bulk-deformation-interpretation would be a surprise, rather than having (in some places in the paper) an opposite point of view where they present the lack of a bulk-deformation as a surprise. More explicitly, it is not clear why the authors present in (5.1) a guess for the deformation in the form of a specific bulk interaction, even though we know precisely what needs to be done on the boundary to reproduce the desired correlation functions, following Witten's [19] and related papers; the simplest way to present the deformation is as a change in the boundary conditions of the 3 fields involved; perhaps it can also be written in terms of boundary interactions, but there is no need to guess their form (since it is determined by the desired modification in the boundary conditions). Of course, beyond the perturbative expansion in the coupling, one can conjecture that there would be an alternative description in terms of bulk physics (as the authors state in the conclusions about the finite coupling theory), but not in perturbation theory in the coupling. I hope that the authors can clarify this conceptual issue (which does not change any of the computations in the paper, but which affects their interpretation).

In addition to this main issue, there are a few more minor issues that I recommend that the authors clarify before the paper is published:

1) In several papers the authors say that obtaining a strongly coupled theory through bulk deformations requires going infinitely far away on the conformal manifold; while this is true for strictly infinite $N$ (when the bulk theory is free in any case), for finite $N$ it seems to require only distances of order $\log(N)$ (as the authors mention at the end of section 2). I think a clearer presentation would be to say that getting strong coupling requires going to very large distances on the moduli space (that diverge at large $N$) or something similar. Infinite distance is required (even at finite $N$) in order to go to points where (say) some sectors of the dual CFT become free, but in order to get strong coupling in the bulk the distance is less than infinite, and this should be clarified.

2) The authors focus on a simple bulk theory in which the $N$-scaling of all the interactions are the same. This is natural to do since it simplifies the analysis, but it may be worth mentioning that in many simple examples of AdS/CFT there are sectors in the bulk that have different $N$-scalings (for instance, this arises when adding to $AdS_5\times S^5$ an orientifold 7-plane with D7-branes on top of it). The authors mention in the discussion that adding branes to the bulk could lead to strongly coupled physics, but they suggest that such branes are disconnected from the standard theories. However, this is not true; for instance, if you consider string theory on $AdS_5\times S^5$ with $g_s$ of order one (which is certainly a standard holographic theory), then going on the moduli space of the CFT involves moving D3-branes in the radial direction, and if several such D3-branes are on top of each other, one obtains strongly coupled physics in the bulk, in a way that is smoothly connected to the standard holographic background. So, I think the authors should do more to clarify that the assumptions behind their analysis are not always valid (even in typical holographic backgrounds).

3) In two places the authors mention that adding a bulk coupling for operators whose dimensions add up to $d$ leads to a divergent correlation function. However, the correct interpretation for such a bulk coupling has been known for many years and it was shown in 1501.06664 that it should be interpreted as giving a beta function for the multi-trace operator (breaking conformal invariance), rather than as giving divergent correlation functions. (The authors mention other options in footnote 15, but I believe this option is the only consistent one, and it was described also in the revised version of [59] which is quoted in this footnote.)

4) The authors should clarify when they discuss holographic theories whether they mean just weakly coupled bulk theories, or if they also mean theories that are local down to distances much smaller than the AdS radius (as in weakly coupled string theories whose string length is much smaller than the AdS radius). When they say in section 3 that symmetric products have been argued to be holographic, I believe that they mean the former sense (not necessarily with locality below the AdS radius), since I don't think one can show that the higher spin currents have very large dimensions (rather than anomalous dimensions of order one). However, when they look for bulk-point singularities, it seems that they mean the latter sense, since I'm not sure if such singularities are expected for theories that are non-local at the AdS radius scale. This issue should also be clarified.

After all these issues are considered and clarified, I will be happy to recommend the publication of this paper in SciPost.

Recommendation

Ask for minor revision

  • validity: high
  • significance: good
  • originality: high
  • clarity: good
  • formatting: perfect
  • grammar: perfect

Author:  Suzanne Bintanja  on 2025-10-31  [id 5972]

(in reply to Report 1 on 2025-08-15)

We thank the referee for the comments and we explain below how we address them.

To address the main issue we reworded the expectation and interpretation in various places. Notably, in the paragraph below eq. (1.5), we reflect the fact that the expectation is that the deformations we consider do not in fact change local bulk couplings. Moreover, we remove the wording in terms of “one possibility” in section 4 in favor of “Following [19], we expect that the OPE coefficients described in the previous section are the result of an interaction that is localized on the boundary,”.

With regards to the minor issues:

  1. We agree that the wording can be improved by specifying that the distance diverges with N in the large N limit. This is now reflected in footnote 4 and in the discussion in the paragraph “Conformal manifolds and infinite distances”.

  2. We specify in the introduction that we start with a CFT that completely factorizes in the large-N limit, and added a footnote regarding the example of AdS5xS5 with D3 branes. Finally, we adjusted the wording towards the end of the paragraph “Conformal manifolds and infinite distances” in the discussion.

  3. We thank the referee for bringing this to our attention, we were previously unaware of this interpretation. We added this interpretation to the introduction, and added a reference to 1501.06664, and promoted footnote 15 to the main text, reflecting the interpretation in terms of an anomaly.

  4. With holographic CFT we always meant to refer to CFTs that have a dual description that at low energies is well described by semiclassical (super) gravity, we thank the referee for noticing the inconsistency in section 3, and rephrased it as: These CFTs have been argued to be connected to holographic CFTs through marginal deformations that reach a holographic point somewhere on their conformal manifold.

We hope that by making these changes to the manuscript we have sufficiently addressed the concerns of the referee.

---

## Round 3 · Referee Report · Anonymous (Referee 2) · 2025-9-19

Strengths

The paper is very well written, clear and contains all the technical details to be able to understand the content.

Weaknesses

Some parts of the paper are a bit speculative.

Report

The paper is very interesting and deals with a very interesting topic.

I would like the authors to clarify a couple of points: 1) It seems to me that the four point function that they found is very simple since essentially one block is the crossing symmetric counterpart of the block corresponding to the other operator. Is this choice unique? Why is it not possible to add another crossing symmetric combination of blocks (infinitely many of them)? This could be interesting also at the level of the bulk point singularity mentioned. 2) If one considers Virasoro symmetry, in which Virasoro multiplet the operators corresponding to the 2 blocks belong?

When these comments are addressed, the paper can be published.

Recommendation

Publish (easily meets expectations and criteria for this Journal; among top 50%)

  • validity: high
  • significance: high
  • originality: top
  • clarity: top
  • formatting: excellent
  • grammar: perfect

Author:  Suzanne Bintanja  on 2025-10-31  [id 5973]

(in reply to Report 2 on 2025-09-19)

Response to referee 2:

We thank the referee for the comments and questions. Below are our responses to the questions posed:

  1. The four-point function we find (written in the introduction in (1.8) and later in (4.7)) is the result of a calculation in conformal perturbation theory, performed in section 4, and is therefore unique. The calculation shows that the four-point function consists of two global conformal blocks, and hence, this decomposition is unique (in the s-channel). One could choose to write it instead in a different channel, but the different expressions would simply be related by crossing symmetry. Of course, the answer that we find is not the unique four-point function that is crossing invariant, and what the referee suggests can be done at the level of crossing symmetry. But it would not correspond to the theory that we are studying.

  2. The two blocks that appear in the resulting four-point function are global blocks associated to the SUSY descendant of the chiral primary O (which we call chi), and the (double-trace) chiral primary :OO:. Each of them corresponds to a Virasoro primary, but the rest of the Virasoro block is suppressed by powers of 1/N. In the limit of N to infinity, the Virasoro block has become the global block. At finite N, the rest of the Virasoro block is there of course, and it would be interesting to understand how to extract it from conformal perturbation theory.

---

## Round 4 · Referee Report · Anonymous (Referee 1) · 2025-11-5

Report

Following the changes and clarifications made by the authors that address the issues raised in my first report, I am happy to recommend the publication of this paper.

Recommendation

Publish (easily meets expectations and criteria for this Journal; among top 50%)

---

## Round 4 · Referee Report · Anonymous (Referee 2) · 2025-11-14

Report

I would like to thank the authors for the replies to my comments and I am happy to suggest the publication of the paper as it is.

Recommendation

Publish (easily meets expectations and criteria for this Journal; among top 50%)

---

## Round 4 · Author Response

This new submission addresses the comments received by the referees as outlined in the list of changes and the replies to the referee reports.

---

## Round 4 · List of Changes

Added a footnote at eq. (1.5) clarifying the assumption that we start with a completely factorizing CFT

Rephrased the paragraph below eq. (1.5) explaining the expectation for the bulk interpretation of the deformed CFT

Replaced infinite distance with distance that diverges with N in footnote 5 and the paragraph “Conformal manifolds and infinite distances” in the discussion section

Added the interpretation of a bulk Witten diagram with cubic vertices whose dual operator dimensions add up to d in terms of an anomaly on page 4 and promoted footnote 15 that addressed the interpretation in the previous version to the main text

Clarified that symmetric product orbifolds are not holographic, but have been argued to be connected to holographic CFTs through marginal deformations that reach a holographic point somewhere on their conformal manifolds in the first paragraph of section 3

Replaced the phrasing of "One possibility" with "Following [20] and related work, we expect that" in the second paragraph of section 4

Rephrased the sentences around eq. (5.1) as an expectation for nature of the bulk theory, and explaining its interpretation as implementing a change in boundary conditions for the bulk fields

Rephrased the final sentence of the paragraph “Conformal manifolds and infinite distances” in the discussion section regarding the addition of branes to obtain strongly coupled matter

---

## Editorial Decision

published